# Antiparasitic Drugs against SARS-CoV-2: A Comprehensive Literature Survey

**DOI:** 10.3390/microorganisms10071284

**Published:** 2022-06-24

**Authors:** Estefanía Calvo-Alvarez, Maria Dolci, Federica Perego, Lucia Signorini, Silvia Parapini, Sarah D’Alessandro, Luca Denti, Nicoletta Basilico, Donatella Taramelli, Pasquale Ferrante, Serena Delbue

**Affiliations:** 1Department of Biomedical, Surgical and Dental Sciences, University of Milan, 20122 Milan, Italy; maria.dolci@unimi.it (M.D.); federica.perego@unimi.it (F.P.); lucia.signorini@unimi.it (L.S.); luca.denti@unimi.it (L.D.); nicoletta.basilico@unimi.it (N.B.); pasquale.ferrante@unimi.it (P.F.); serena.delbue@unimi.it (S.D.); 2Department of Biomedical Sciences for Health, University of Milan, 20133 Milan, Italy; silvia.parapini@unimi.it; 3Department of Pharmacological and Biomolecular Sciences, University of Milan, 20133 Milan, Italy; sarah.dalessandro@unimi.it (S.D.); donatella.taramelli@unimi.it (D.T.)

**Keywords:** COVID-19, SARS-CoV-2, antimalarials, anthelmintics, antiparasitics, drug repurposing

## Abstract

More than two years have passed since the viral outbreak that led to the novel infectious respiratory disease COVID-19, caused by the SARS-CoV-2 coronavirus. Since then, the urgency for effective treatments resulted in unprecedented efforts to develop new vaccines and to accelerate the drug discovery pipeline, mainly through the repurposing of well-known compounds with broad antiviral effects. In particular, antiparasitic drugs historically used against human infections due to protozoa or helminth parasites have entered the main stage as a miracle cure in the fight against SARS-CoV-2. Despite having demonstrated promising anti-SARS-CoV-2 activities in vitro, conflicting results have made their translation into clinical practice more difficult than expected. Since many studies involving antiparasitic drugs are currently under investigation, the window of opportunity might be not closed yet. Here, we will review the (controversial) journey of these old antiparasitic drugs to combat the human infection caused by the novel coronavirus SARS-CoV-2.

## 1. Introduction

Considering that the COVID-19 pandemic to date has caused more than 6 million deaths worldwide, increasing the number of specific and effective therapeutic agents targeting SARS-CoV-2 represents an urgent and unmet need.

SARS-CoV-2 is an enveloped virus with a single stranded RNA of positive polarity, 29.9 kb in length, containing non-structural (nsps) and structural/accessory proteins [1,2]. The virus entry into host cells is mediated by the binding between the Receptor Binding Domain (RBD) of the viral structural Spike (S) protein and the Angiotensin Converting Enzyme II (ACE2) receptor on the target cells [3]. Following entry, the viral genome is released into the host cytoplasm, where two open reading frames are immediately translated into two polyproteins that are subsequently cleaved, leading to the production of the nsps. Instead, the structural/accessory proteins are synthesized through the translation of subgenomic RNAs (sgRNAs) [1,4]. In particular, viral structural proteins such as membrane (M), envelope (E) and S glycoproteins that are embedded into the viral envelope, are translated and translocated into the endoplasmic reticulum. Upon the replication of the viral genome, new single stranded RNAs of positive polarity are then incorporated into new viral particles [4]. The SARS-CoV-2 life cycle ends when the mature new virions leave the host cells through the exocytic pathway [5].

Concerning the pathogenesis of COVID-19, it greatly varies from patient to patient: most infected people develop no or mild symptoms, such as fever, cough, fatigue, dyspnea, myalgia, sputum production and headache [6], while others show severe symptoms [7]. In the latter patients, viral replication in lung parenchyma may cause severe pneumonia, with organ function damage and severe interstitial inflammation [8], which may induce an abundant production of pro-inflammatory cytokines and chemokines [7], a phenomenon called “cytokine storm” [9]. Resulting symptoms may be due to the cytokine-induced tissue damage or may result from immune-cell–mediated responses [10].

While a variety of drugs targeting many of the SARS-CoV-2 proteins have been evaluated, still no effective antiviral strategy for COVID-19 exists to date [11]. Currently, the few drugs for COVID-19 treatment that have gained full approval from the US Food and Drug Administration (FDA), or an emergency authorization, are mainly represented by repurposed agents: the antiviral remdesivir (fully approved), the antivirals Paxlovid (nirmatrelvir/ritonavir combination) and Lagevrio (molnupiravir), and the monoclonal antibodies Evusheld, Tocilizumab, and Bebtelovimab (authorized in case of emergency).

However, especially during the first period of the pandemic, when medicine seemed completely unable to find tools to cure COVID-19 or at least to reduce its high level of lethality, some antimalarial drugs with reported antiviral activities were used to fight SARS-CoV-2 [12]. Indeed, the repurposing approach, that is the usage of existing drugs for new therapeutic indications, is a well-known and applied strategy against epidemic virus, to reduce development costs, time to market, and risks of failure [13]. Hence, during the last SARS-CoV-2 pandemic, drug repurposing screens have emerged as an attractive strategy to accelerate new drug discovery and development pipelines, by focusing the attention on known drugs with direct/indirect antiviral effects [14].

Few months before the pandemic occurrence, we reviewed the use of antimalarial drugs against all known viral infections [15], concluding that their repurposing might be useful, especially in cases of antiviral drug-resistance or during the emergence of new viruses for which effective drugs were not readily available.

The aim of the present study was to review the available data on well-known antiparasitic agents with a special emphasis on antimalarial and anthelmintic drugs, given their significant potential to be repurposed as anti-SARS-CoV-2 drugs (Figure 1). The literature screen has been performed using the PubMed search engine by searching “SARS-CoV-2” or “COVID-19” and the single name of specific antiparasitic agents. Current knowledge of the efficacy of these drugs by in vitro/in silico, in vivo studies and clinical trials, along with their proposed mechanisms of action (MoA) against SARS-CoV-2 (Figure 2), will be discussed throughout the different sections of the present review.

## 2. Antimalarial Drugs

### 2.1. The Controversial Journey of 4-Aminoquinolines: Chloroquine (CQ) and Hydroxychloroquine (HCQ)

In 1880, the French military doctor Alphonse Laveran discovered the etiologic agent of malaria, a parasitic protozoan infecting the blood cells of the human host [16]. Already by that time, quinine, a natural compound derived from the bark of the cinchona tree, was widely used to treat malaria-infected patients [17]. It was not until 1934, that a 4-aminoquinoline termed chloroquine (CQ) (Figure 1) became the first synthetic quinine substitute and the drug of choice against malaria [18]. Later in 1955, continued chemical modification of CQ resulted in the introduction of its hydroxy-analogue hydroxychloroquine (HCQ) (Figure 1), which rapidly proved to be 3-fold less toxic and more water soluble [19].

Beyond their antimalarial action, CQ/HCQ exhibit anti-infective, anti-inflammatory, immunomodulating and antithrombotic therapeutic properties, resulting from not fully characterized mechanisms [20]. These pleiotropic abilities rapidly prompted the repurposing of both aminoquinolines as a potential treatment of several non-malarial diseases, ranging from infectious, rheumatological and autoimmune to neurological conditions [21].

As anti-infectives, CQ/HCQ have been successfully used to treat bacterial [22,23], and fungal infections [24,25,26]. Even though their exact antiviral mechanism of action is still unclear, CQ/HCQ may abrogate viral entry and replication inside host cells by altering endosomal acidification and pH-dependent enzymatic cleavage, by inhibiting post-translational modifications of viral proteins, or by restricting cellular iron accumulation [27,28,29]. Additionally, CQ/HCQ exert indirect immunomodulatory activities by suppressing the production of pro-inflammatory cytokines and immune hyperactivation (Figure 2) [30]. Notably, the effectiveness of CQ/HCQ analogs has been investigated in emerging and re-emerging viruses such as HIV [31,32], dengue and hepatitis C [33,34], human and avian influenza [35,36], Chikungunya virus (CHIKV) [37], Ebola and Marburg viruses [38,39], and severe acute respiratory syndrome (SARS) and Middle-East respiratory syndrome (MERS) coronaviruses [40,41], all of them representing huge challenges to human and veterinary medicine nowadays.

The current fight against SARS-CoV-2 has not been an exception, and the repositioning of CQ and HCQ as promising weapons to combat COVID-19 was rapidly investigated, partially supported by the enormous experience of CQ/HCQ use in malaria chemoprophylaxis, excellent safety and tolerability profile, very low cost and their in vitro broad-antiviral properties.

#### 2.1.1. CQ and HCQ against SARS-CoV-2: In Silico and In Vitro Studies

Shortly after the COVID-19 outbreak and in the absence of any available pharmacological treatment and/or vaccine, one of the first published reports showed that SARS-CoV-2 infection in the common African green monkey kidney-derived Vero E6 was potently impaired by CQ treatment at low micromolar concentrations, and also suggested that therapeutic doses might be clinically achievable [42]. The same group found that HCQ was also effective in inhibiting SARS-CoV-2 infection in vitro, and more potently blocked the transport of virions into late endolysosomes (membrane-bound vesicles in the endocytic pathway) (Figure 2) [43], which was rapidly corroborated by another report [44].

Contrarily, in July 2020, a study by Hoffmann et al. compared the inhibition by CQ and HCQ of SARS-CoV-2 entry into Vero cells, Vero cells expressing TMPRSS2 (a serine protease that activates SARS-CoV-2 for entry into lung cells [45] and is essential for MERS-CoV pathogenesis in infected murine models [46]), and Calu-3 cells, airway epithelial cells that naturally express TMPRSS2 [45]. The report identified that CQ/HCQ failed to block infection with SARS-CoV-2 in TMPRSS2-Vero and Calu-3, indicating both an unlikely antiviral activity in human lung tissue and protection against COVID-19 [47].

Since CQ and HCQ are weak bases that preferentially accumulate in acidic organelles (endolysosomes and Golgi apparatus) [48], both elevate the pH of the acidic lumen and prevent the release of the viral genome into the host cytoplasm (Figure 2), as previously seen for human and avian influenza A viruses [49]. Similarly, the anti-SARS-CoV-2 lysosomotropic effects of CQ were confirmed in two human cell lines including Huh-7 (human hepatocarcinoma cells) and 293T-hACE2 (human kidney cells), showing that CQ (40 μM) markedly reduced viral replication [50]. Conversely, another report found that increasing concentrations of CQ (10, 50 and 100 μM) minimally altered the endosomal pH of human gastric epithelial cells [51]. Moreover, CQ induced Golgi-deacidification and affected post-translational modifications including glycosylation of SARS-CoV virions in vitro [40]. It is possible that another not-yet-confirmed mechanism of CQ/HCQ against SARS-CoV-2 might include the impairment of post-translational glycosylation of viral proteins, such as the Spike. Besides, CQ/HCQ can also block the uptake of SARS-CoV-2 virions by affecting the glycosylation of the ACE2 receptor in the plasma membrane (Figure 2) [43].

Considering that CQ reduces the expression of phosphatidylinositol binding clathrin assembly protein (PICALM), a clathrin adaptor essential for clathrin-mediated endocytosis [52] and blocks the endocytosis of nanoparticles in a clathrin-dependent manner [53,54], it is thus tempting to speculate that CQ might also interfere with this process during SARS-CoV-2 entry into host cells (Figure 2). However, clear evidence is still missing.

To speed up the anti-SARS-CoV-2 drug discovery pipeline, in silico bioinformatic molecular docking has been applied to find undisclosed CQ/HCQ cellular targets and to assess the binding affinities of both drugs with viral and host proteins, including: the viral protease (M^Pro^) and host cathepsin L (CTSL), as the main proteolytic systems involved in the viral Spike protein activation [55,56]; and the RBD of the Spike protein, allowing virus entry and replication in host cells [45,57]. Results showed that HCQ achieved better interactions and affinity with ACE2 and M^Pro^, whilst CQ achieved better results with M^Pro^ and CTSL, probably due to structural differences between the two drugs (Figure 2) [58]. In addition, Amin et al. conducted an in silico study and further demonstrated that HCQ exhibited improved binding affinities than CQ to the viral NTD-N-protein [59].

Furthermore, as a β-coronavirus, SARS-CoV-2 relies on glycoproteins and sialylated gangliosides as main attack sites of the respiratory epithelium [60]. Through molecular and structural modelling, Fantini et al. identified a ganglioside-binding site (GBS) in the N-terminal domain of the Spike glycoprotein, that may improve viral attachment to lipid rafts facilitating the interaction with the ACE2 receptor [61]. Interestingly, CQ and HCQ emerged as potential blockers of the initial S–ganglioside binding to the surface of respiratory cells (Figure 2), supporting the use of both drugs as initial therapy in infected individuals [61]. Remarkably, through in silico combination of drug-gene interaction networks, molecular docking and virus-host-drug interactome mapping [62], an unknown strong binding affinity of CQ/HCQ to TLR9 and IL-6 (components of innate immune response) was seen. Moreover, TLR9 and IL-6 were identified as targets of several SARS-CoV-2 proteins (Figure 2) [62]. Given that inhibition of TLR signaling is a promising option for COVID-19 therapeutics [63], the use of CQ/HCQ as TLR-dependent inhibitors is of significant importance.

Interestingly, in silico docking and dynamics studies have recently identified ACE2 receptor as well as novel HCQ targets including the α7 nicotinic AcetylCholine Receptor (α7 nAChR), α1D-adrenergic receptor (α1D-AR), Histamine N-Methyl Transferase (HNMT) and DNA gyrase/Topoisomerase III β (Top3β) [64]. In particular, the authors found that HCQ would block virus-binding sites on both ACE2 and α7 nAChR at the entry stage, whereas at post-entry stages, HCQ would prevent viral replication by acting against Top3β, and the “cytokine storm” by inhibiting α1D-AR.

In January 2021, Doharey et al. docking analyses identified CQ/HCQ as direct inhibitors of SARS-CoV-2 RNA-dependent RNA polymerase (RdRp), a key viral protein for RNA replication and transcription, and thus an important target for the development of antiviral drugs (Figure 2) [65]. In this regard, the authors found that CQ/HCQ showed the highest affinity towards SARS-CoV-2-RdRp, binding to its active site in the same manner as its substrate ATP binds [65].

Overall, in silico computational analyses have highlighted the potential of CQ and HCQ to interfere with SARS-CoV-2 infection at different layers, and support additional studies on the use of both drugs against COVID-19.

#### 2.1.2. Preclinical In Vivo Use of CQ and HCQ

On 28 March 2020, the urgent need for an effective treatment against COVID-19 prompted the FDA to issue an emergency authorization to prescribe CQ and HCQ for COVID-19 patients despite the contrasting in vitro data and lack of preclinical evidence of their in vivo efficacy [66].

In May 2020, interesting data were reported on the therapeutic potential of HCQ in a ferret infection model [67], which recapitulates aspects of human SARS-CoV-2 infection and transmission [68]. In particular, HCQ-treated ferrets exhibited lower clinical scores compared to non-treated animals, whereas virus titers in nasal washes, stool specimens and respiratory tissues were similar among both groups. Two months later, the antiviral efficacy of HCQ was evaluated in SARS-CoV-2-infected cynomolgus macaques [69], a relevant model for the analysis of the early stages of SARS-CoV-2 infection in humans [70]. Differently from the in vitro data, HCQ had no significant effect on viral load, despite the high concentrations in the blood and lungs of infected animals [69]. Even more, when the drug was used as a prophylactic treatment, HCQ did not confer protection against SARS-CoV-2 infection. Consistent with these findings, in October 2020, standard human malaria HCQ prophylaxis and treatment did not benefit clinical outcome nor reduce SARS-CoV-2 titers in the upper and respiratory tract in the rhesus macaque disease model and in infected Syrian hamsters [71]. Moreover, Kaptein et al. added more negative results about the inexistent antiviral activity of HCQ in SARS-CoV-2-infected hamsters [72], suggesting no scientific basis for the use of HCQ in COVID-19 patients [72]. More recently, it was reported that HCQ and azithromycin (AZ, an antibiotic with broad-spectrum antiviral activity), alone or in combination, did not block SARS-CoV-2 replication in primary human bronchial airway epithelia, displaying no significant effect on viral replication, clinical course and lung impairment when tested in the Syrian hamster model [73].

Concerning CQ, a study searching for inhibitors of endosomal acidification to suppress SARS-CoV-2 replication and infection in vivo, showed that CQ reduced viral replication in the lungs and alleviated pneumonia reducing inflammation and infiltration in hACE2 transgenic mice [50,74].

Overall, only few reports have assessed the preclinical efficacy of CQ/HCQ against COVID-19 so far. It is thus tempting to speculate that in the absence of such an urgency of therapeutic options and vaccines against COVID-19, more preclinical studies would have possibly warned against the use of CQ/HCQ during SARS-CoV-2 infection in vivo, likely discouraging its further testing in human clinical studies.

#### 2.1.3. CQ and HCQ in Clinical Trials

As early as in March 2020, 15 clinical trials had been already registered in the Chinese Clinical Trials Registry (ChiCRT) to test the efficacy and safety of CQ and HCQ in the treatment of COVID-19 associated pneumonia [75]. First published results from those multicenter trials showed apparent efficacy and acceptable safety against COVID-19, which encouraged the authors to further recommend the use of CQ/HCQ to treat larger infected populations [75].

In contrast, a randomized CloroCOVID-19 clinical trial aiming at comparing high vs. low dosage of CQ showed more cardiac toxicity and lethality in the high dosage arm, indicating that such dosage should be avoided for the treatment of severe COVID-19 [76]. Subsequently, in the attempt to assess the safety and benefit of the administration of CQ/HCQ in combination with AZ, an international, observational registry of 96,032 COVID-19 patients, found that the use of CQ/HCQ in different regimens was associated with ventricular arrythmia and an increased hospital death rate with no evidence of benefit of their use. Amazingly, despite unproven efficacy, CQ/HCQ were rapidly praised as a potential miracle cure for COVID-19 in the early days of the pandemic by many leaders including the US ex-president Donald Trump, circumstances that further pushed its use across the globe [77]. Indeed, by May–June 2020, the negative findings reported by Mehra et al. stopped many clinical trials in their tracks. However, on 4 June (only two weeks after its initial publication), sudden concerns were raised about the reliability of the data and analyses conducted by Surgisphere Corporation, the healthcare analytics company involved in the study, forcing the authors to request the retraction of the article from The Lancet.

On 5 June 2020, a press release announced the results from one of the largest trials, the RECOVERY randomized clinical trial on HCQ, by stating that any clinical benefit was found in hospitalized patients with COVID-19, and that the HCQ arm of the study was immediately stopped [78]. In contrast, in July 2020, Gautret and collaborators published the results from a small open-label non-randomized clinical trial which showed that HCQ treatment was significantly associated with viral load reduction in COVID-19 patients, and a synergistic and positive effect of HCQ combined with AZ [79].

Moreover, HCQ failed to prevent illness compatible with COVID-19 or confirmed infection when used as high-risk or moderate-risk post exposure prophylaxis within 4 days after exposure [80]. In addition, Mitjà et al. reported the lack of efficacy of HCQ to prevent SARS-CoV-2 infection or symptomatic COVID-19 disease in healthy persons exposed to a PCR-positive case patient [81].

Conflicting and insufficient data on the effect of CQ/HCQ in terms of all-cause mortality, progression to severe disease, clinical symptoms, and upper respiratory virologic clearance with antigen testing are available as well [82,83]. From the contrasting findings published in February 2021 by the SOLIDARITY Trial Consortium, the World Health Organization concluded that HCQ has little or no effect on hospitalized COVID-19 patients according to mortality, initiation of ventilation and duration of hospital stay [84]. Besides, the lack of efficacy of HCQ and HCQ/AZ for outpatient treatment [85], the absence of benefit of early treatment with HCQ for decreasing hospitalization [86], or the increased risk of major cardiovascular events with no effects on viral clearance rates, were reported in outpatients with early and mild COVID-19 disease [87]. More recently, in March 2022, a double-blind, multicenter, randomized, controlled trial attempted to increase the statistical power by including 1372 patients that were randomly allocated to HCQ or placebo [88], and further showed that in outpatients with mild/moderate forms of COVID-19, treatment with HCQ did not reduce the risk of hospitalization [89].

Although not always properly investigated, the use CQ/HCQ as possible treatments against COVID-19 would require a careful examination of their pharmacokinetic properties, safety profile and potential drug interactions. Despite being relatively safe in the treatment of malaria and rheumatic diseases with standardized doses, CQ/HCQ can lead to dermatological changes or adverse reactions of the gastrointestinal tract (i.e., nausea, vomiting, diarrhea), whereas the most severe side effects may include cardiotoxicity, neuromyopathy of proximal muscles, and irreversible retinopathy, mainly during prolonged treatment or high doses [90]. Although rare, case reports have also described severe hypoglycemia in malaria patients treated with CQ/HCQ, as well as in individuals with lupus and other chronic diseases [91]. Both drugs are quickly and fully absorbed after oral administration, and about 50–60% of the CQ/HCQ in blood is bound to proteins [92]. After being metabolized in the liver, high drug concentrations are found in the cardiac tissue, lungs, kidneys, liver, skeletal muscle, skin, and the eye [93]. Indeed, CQ/HCQ pharmacologically active metabolites have been involved in the drug-induced cardiotoxicity [94] and pruritus, but to date no data are available on their performance in COVID-19. Concerning cardiotoxicity, several of the studies reported above reported the risk of serious cardiovascular adverse effects (i.e., QTc prolongation, ventricular arrhythmia, atrial fibrillation and cardiac arrest) in COVID-19 patients treated with HCQ/CQ monotherapy or HCQ/CQ + AZ, which highlights the real need of evaluating the potential benefit/harm balance of CQ/HCQ in the treatment of SARS-CoV-2 infection [95]. Moreover, pharmacodynamic drug–drug interactions (DDIs) of CQ/HCQ with other medications must also be taken into consideration. For example, co-administration of HCQ with several antiviral agents including lopinavir-ritonavir, darunavir-cobicistat, and acetazolamide, resulted in QT-interval prolongation, ventricular arrhythmias, and torsade de pointes [96]. In addition, interference of HCQ with darunavir-cobicistat and tocilizumab might also lead to psychiatric disorders, such as behavioral disturbances, psychosis, agitation, delirium, and aggression [96]. Moreover, in COVID-19 patients, the administration of HCQ with acetazolamide and the anticancer ibrutinib resulted in a tachyarrhythmia as DDI [97]. Given that around 12% of hospitalizations in oncology units are due to adverse effects of anticancer agents [98], potential drug interactions with CQ/HCQ should be cautiously monitored.

Currently, 96 clinical studies are registered to ClinicalTrials.gov with the terms “chloroquine” and “COVID-19”, 35 of them already active, while HCQ accounts for a total of 291 registered trials and 92 studies under investigation (Appendix A Appendix A) (accessed on 3 May 2022). The main clinical studies involving CQ and HCQ are summarized in Table 1.

### 2.2. Quinine and other Aryl-Aminoalcohols: In Vitro and In Vivo Studies

Aryl-aminoalcohol is a chemotype present in potent antimalarial drugs, including quinine (Q), mefloquine (MQ), lumefantrine, and halofantrine (Figure 1). Beyond their antimalarial activity, the antiviral effects of these molecules have been previously studied (reviewed in D’Alessandro et al. [15]).

Indeed, numerous manuscripts have been recently published on the possible anti-SARS-CoV-2 activity of these molecules, with a particular focus on Q and MQ. Although there is in vitro evidence that Q restricts SARS-CoV-2 infection, neither clinical nor preclinical data have been reported so far [102]. Latarissa et al. recently published a review on this topic [103]. Briefly, in silico studies indicate several possible MoA of Q, including the binding to the Lys353 residue in the peptide domain of the ACE2 [104], to the viral non-structural protein nsp12 [105], and to the SARS-CoV-2 main protease and the Spike glycoprotein (Figure 2) [106]. These computational binding affinities have been demonstrated in vitro, showing that Q exerts its antiviral activity against SARS-CoV-2 in several cell lines with IC_50_ values ranging from 10 μM to 60 μM [107,108].

Both in silico and in vitro tests [109,110] have been carried out to evaluate MQ efficacy against SARS-CoV-2, but studies reporting on in vivo infection models or clinical trials are still missing. Interestingly, all reports confirmed a good antiviral activity on different cell lines, although few studies attempted to understand its MoA. The most used cell line has been the Vero E6 with an IC_50_ ranging from 1.8 μM to 8.06 μM [111,112,113,114,115], and similar results were obtained on Caco-2 [116] and Calu cells [112] as well. MQ MoA seems to interfere with the viral load, resulting in a reduction of the cytopathic effects in vitro. An in silico study demonstrated that MQ and halofantrine bind effectively to the M^Pro^ [110]. Besides, a time-of-addition analysis performed in Vero E6/TMPRSS2 cells indicated that MQ inhibited viral entry early after viral attachment to the target cell, whereas a lower antiviral effect was observed post-entry [112]. Finally, the antiviral activity of MQ appears to be enhanced in combination with other antimalarials or with viral replication inhibitors [112,117]. MQ is usually well tolerated, but in people with active liver or thyroid disease it can cause adverse effects in the nervous and gastrointestinal systems [118,119].

Overall, despite numerous studies investigating the anti-SARS-CoV-2 effect of aryl-aminoalcohols, much work is still needed to confirm their beneficial effects against COVID-19 infection. Furthermore, the lack of in vivo and clinical trials likely suggests a difficult translation from in vitro to in vivo studies.

### 2.3. Artemisinin and Its Derivatives

Artemisinin (ART), also known as quinghao, is a sesquiterpene lactone with a unique peroxide structure deriving from the Chinese medicinal plant *Artemisia annua* L. (Figure 1) [120]. Several semi-synthetic derivative drugs have been obtained from ART: artemether, artemisone, artesunate (AS), artefenomel (OZ439) and artenimol or dihydroartemisinin (i.e., the active metabolite of ART). These drugs have been widely repurposed to treat several diseases including viral infections [121,122,123].

#### 2.3.1. Artemisinin: In Vitro Studies

Several in silico studies have been performed to define the targets of ART both on the pathogen SARS-CoV-2 and on the host sides. Excellent affinities of ART have been demonstrated with both a close and an open conformation of the Spike glycoprotein and with one South African variant [124], with the RBD of the Spike (Figure 2) [125]. However, the energy of bond of ART to the Spike was weaker compared to other antimalarial drugs such as CQ or MQ [126,127]. Concerning M^Pro^ and thus a potential arrest of the viral replication (Figure 2), the variable binding energies obtained indicate that M^Pro^ is not an ART target [127,128,129,130,131,132], although the hybridization of ART with thymoquinone resulted in an amelioration of the affinity with the viral protease [133]. Additionally, relatively good affinities of ART to the papain-like protease [127] and the non-structural protein 1 (nsp1) [134] have also been reported (Figure 2).

Among the host’s targets, ART was found to discretely interact with ACE2, TMPRSS2 and neuropilin-1 receptors, and the Glucose Regulated Protein 78 (GRP78, an endoplasmic reticulum protein that translocates to the plasma membrane and facilitates viral entry through binding to the substrate binding domain of SARS-CoV-2) [124,132]. The ART-ACE2 interaction which appears to be strongly influenced by the polymorphisms of the host receptors, has also lowered its priority use [128].

Successive in vitro studies further showed that increasing doses of ART exhibited a reduced ability to prevent SARS-CoV-2 pseudovirus entry compared to that of HCQ on the HEK293T/ACE2 cell line [130], and its anti-SARS-CoV-2 activity was proved to be lower compared to Artemether and AS in a plaque-reduction assay in Vero E6 cells [135].

Notably, to mitigate the “cytokine storm” in COVID-19 patients, the immunosuppressive effects of ART proved to reduce the production of pro-inflammatory TNFα both from SARS-CoV-2 pseudovirus and LPS-treated THP-1 macrophages at 25 and 50 mM, together with a decrease in the production of CXCL8 from THP-1 pre-stimulated with the pseudovirus (Figure 2) [130].

#### 2.3.2. Artemisinin: Clinical Trials

ART has also been tested in few clinical trials. An open label, prospective, multi-center, comparative and interventional study was conducted on 120 patients treated with cycles of 500 mg of ART/day (ARTIVeda™) as a dietary supplement to the different SoC, demonstrating a faster recovery when the dietary supplementation was performed in mild or moderate COVID-19 patients [136]. Details of the above-mentioned clinical trial are shown in Table 2.

#### 2.3.3. Artemisinin Derivatives: Artefenomel (OZ439), Artemether, Artemisone, Artenimol (Dihydroartemisinin, DHA), AS

Beyond ART, the antiviral activity of its derivatives, AS, DHA and artemisone, has also been investigated in in silico studies concluding that only AS and DHA formed stable complexes with the Spike glycoprotein or the RBD (Figure 2) [137]. Instead, DHA, but not artemether, was docked with the RBD, confirming a quite good interaction [110,125]. On the other hand, focusing on inhibiting SARS-CoV-2 replication, several viral nsps were tested as hypothetical targets of the ART derivatives. Unlike AS and artefenomel, which showed a tight and stable interaction with the binding pocket of the M^Pro^ of SARS-CoV-2 (Figure 2) [138], artemether has been demonstrated once again to be the worse ART derivative [110]. Similarly, the computational binding of AS to the non-structural proteins Helicase (or nsp13), nsp10, nsp14 and nsp15 resulted to be much more favorable and stable compared to artemether [109]. Furthermore, a suitable interaction of AS with nsp1 was also found (Figure 2), including a structural topology change upon molecule binding [134]. In the same study AS and artemether were in silico docked to two viral structural proteins, the E and the N proteins, both important for the packaging of the viral genome (Figure 2) [109,139]. In contrast to ART, none of the literature examined for this review presented in silico studies on the interactions with host’s structures.

Side by side to computational in silico approaches, several in vitro investigations have been reported. AS, the best interacting molecule with SARS-CoV-2 proteins from computational studies, showed an EC_50_ ranging from 12.98 mM to 16.24 mM and a selectivity index (SI) from 5.10 to 7.84 calculated by RT-PCR in the supernatants of infected Vero E6 cells after 24 h [140,141]. Very potent activity of AS was also detected using Huh7.5 and A549-hACE2 cells [135]. Moreover, it reduced the expression of the viral N protein in the micromolar range (Figure 2) [140,141], and a time-of-addition assay established a post-entry stage activity by AS [135,141]. However, 30 mM AS failed to inhibit SARS-CoV-2 cytopathic effects on Vero E6 cells at 72 h post-infection and did not reduce virus replication in Calu-3 cells after 48 h of treatment [142]. Even with poor in silico evidence, DHA emerged as a promising antiviral molecule. Like AS, the EC_50_ of DHA by RT-PCR on infected Vero E6 cells after 24 h was 13.31 mM, but with a lower SI of 2.38 [140]. Instead, after 48 h the EC_50_ raised to 20.1 mM [102]. Again, DHA also reduced the viral N expression in the micromolar range (Figure 2) [140]. Likewise, artemether showed a complete inhibition of N expression at 6.25 mM, functioning both at entry and post-entry stages [141]. Moreover, the EC_50_ varied between 53 and 98 mg/mL on different cell types, including Vero E6, Huh7.5, A549-hACE2 lines [135]. Only one study investigated the in vitro efficacy of 30 μM OZ439 demonstrating a 1.5 log decrease of SARS-CoV-2 RNA load in Vero E6 cells, and a 1 log decrease in Calu-3 cells. Nevertheless, the discrete toxicity revealed by biomass analysis in Calu-3 discouraged its use [142].

#### 2.3.4. AS: Clinical Trials

Among all the artemisinin derivatives, only two clinical trials evaluating the in vivo efficacy of AS are ongoing. In a Phase II randomized and double-blind clinical trial, 100 mg of AS administered once a day for 5 days will be tested and compared to placebo for its effects in reducing the length of hospital stay, and in shortening the time needed for a COVID-19 test to become negative (Appendix A Appendix A). Differently, the effects of AS injections 2.4 mg/kg/dose in addiction to local SoC on the hospital length of stay and mortality will be assessed in a phase IV, randomized and open label clinical trial (Appendix A Appendix A).

#### 2.3.5. Artemisinin and Its Derivatives in Combination with Other Molecules: In Vitro Studies

Two in vitro studies and nine clinical trials are currently focused on the use of ART or its derivatives in combination with other drugs, including antimalarials and chemically different compounds.

Gendrot and colleagues assessed the anti-SARS-CoV-2 activity in Vero E6 of different drug combinations (AS—amodiaquine, artemether—lumefantrine, AS—pyronaridine and DHA—piperaquine), and reported the use of AS 250 mg with MQ 550 mg as the best tandem among all drugs tested [117]. Moreover, the same group also evaluated the possible synergistic effect of DHA when combined with methylene blue, which resulted, however, in an antagonistic effect [111].

#### 2.3.6. Artemisinin and Its Derivatives in Combination with Other Molecules: Clinical Trials

Concerning the use of ART alone or in combination with other molecules (Appendix A Appendix A), only one clinical study has been completed to date, although the final results have not been published, yet. By evaluating the effect of an oral spray called ArtemiC (artemisinin, curcumin, frankicense and vitamin C) administered twice/day compared to placebo, excellent safety and efficacy profiles were found. In fact, such orally-administered ART used with the SoC or in combination with both SoC and OT-101 (Trabedersen, an orphan drug), is under investigation in a Phase II randomized and double-blind clinical study.

In addition to ART, the combination therapy of AS with other compounds has been also investigated (Appendix A Appendix A), including the use of AS and Pyramax (pyronaridine-AS) to ameliorate COVID-19 symptoms and reduce the viral load, and the use of pyronaridine-AS and the combination of AS and amodiaquine, also in the commercial form of Artecom^®^. Besides, a Phase III observational cohort study on malaria patients is comparing the effects of the administration of AS with pyronaridine versus another antimalarial combination treatment, artemether-lumefantrine. Finally, AS formulation with amodiaquine called Cospherunate is currently under evaluation when administered with AZ in the presence or absence of phytomedicines to treat symptomatic COVID-19 patients (Appendix A Appendix A).

### 2.4. Other Antimalarial Drugs: Atovaquone

Atovaquone is a hydroxynaphthoquinone synthetized during World War II due to the shortage of quinine. Nowadays it is used in a fixed-dose combination with Proguanil (Malarone) for the treatment of uncomplicated malaria or for prophylaxis in travelers [143]. It is also indicated for the treatment of severe pneumonia due to *Pneumocystis jirovecii* in immunocompromised patients [144]. In silico molecular docking studies have identified different proteins of SARS-CoV-2: the main protease [145], the B chain of the RBM [146], the NTD of the N protein [147] and the papain-like proteinase [148] as the most promising targets showing the lowest binding energies with atovaquone. Moreover, subsequent in vitro tests have showed a strong inhibition of the replication of different SARS-CoV-2 variants (alpha, beta and delta) in Vero E6 and Calu-3 cells [149]. Additionally, atovaquone was able to significantly reduced virus replication and infection in human lung cells, Vero cells and human hepatocytes [150,151]. However, concerning atovaquone’s ability to reduce virus production in airway epithelium cultures, inconsistent results have been obtained [149,150].

There are three registered clinical trials involving the use of atovaquone for COVID-19, but only one has been completed (ID NCT04456153). The results show that the addition of 1500 mg of atovaquone to the SoC twice a day, for 10 days, led to a reduction of SARS-CoV-2 load and to a better outcome in 60 patients, although not statistically significant compared to the placebo group.

## 3. Other Antiprotozoan Drugs: K777

In 2021, a cysteine protease inhibitor termed K777 (also known as SLV213) was reported to block SARS-CoV-2 infection in different human and monkey cell lines in vitro without cytotoxic effects [152]. K777 had previously been shown to be safe and efficacious in mice and dogs infected by *Trypanosoma cruzi*, the protozoan parasite which causes Chagas disease, a leading cause of death in Latin America [153,154]. As an antiparasitic agent, K777 inhibits cruzipain, the major cysteine protease of *T. cruzi* involved in parasite replication, host cell invasion, and subversion of the host immune response [155,156,157]. K777 also selectively inhibits the human cysteine protease cathepsin L (CTSL) (Figure 2), but not its SARS-CoV-2 counterparts papain-like and 3CL-like proteases [152]. CTSL functions by cleaving the SARS-CoV-2 Spike protein which in turn enhances virus entry. Indeed, elevated circulating levels of CTSL have been positively correlated with disease course and severity in COVID-19 patients [56]. Thus, the inhibition of CTSL by K777 may result in the loss of cathepsin L-mediated Spike processing, preventing viral infection. Indeed, K777 had already demonstrated its antiviral effects back in 2015, when Zhou et al. reported that K777 blocked the entry of SARS-CoV-1 and MERS pseudoviruses into Vero E6 and HEK293 cells, likely due to the inactivation of CTSL on cell surfaces and/or within endosomes [158].

A single, double blind, placebo-controlled clinical trial by Selva Therapeutics about the efficacy of K777 on clinical symptoms of COVID-19 is currently ongoing (ClinicalTrials.gov Identifier: NCT04843787).

## 4. Anthelmintic Drugs against SARS-CoV-2

### 4.1. Niclosamide

Niclosamide (NIC; 5-Chloro-N-(2-chloro-4-nitrophenyl)-2-hydroxybenzamide) (Figure 1) was originally developed in 1953 as a molluscicide to kill snails. Following its approval by the FDA in 1982, it was found to be an effective treatment for gastrointestinal parasitic infections caused by tapeworms in humans [159,160]. Its main MoA seems to be the blocking of glucose uptake, thus acting as an uncoupling agent for energy-generating oxidative phosphorylation, starving the worms of ATP and ultimately impacting the parasite’s pH homeostasis [159]. It also acts on other cellular signaling pathways such as Wnt/β-catenin, mTOR and JAK/STAT3. Remarkably, niclosamide has demonstrated broad antiviral effects against a wide number of different viruses such as SARS-CoV, MERS-CoV, Dengue and Zika virus, Japanese encephalitis virus, hepatitis C, Ebola, human rhinoviruses, Chikungunya virus, human adenovirus and Epstein–Barr virus [161,162]. As antiviral, niclosamide exploits the neutralization of the endo-lysosomal pH, which interferes with pH-dependent membrane fusion events, a critical step that restricts virus entry [161].

#### 4.1.1. Niclosamide: In Vitro and In Vivo Studies

The anti-SARS-CoV-2 activity of niclosamide was reported early in July 2020, when a screening of a panel of 48 FDA-approved drugs previously preselected by an assay on SARS-CoV-2 identified niclosamide as one of the most relevant compounds in preventing the infection in Vero cells, exhibiting an IC_50_ value in the low micromolar range (0.28 µM) [113]. Against SARS-CoV-2, two main MoA have been considered: (i) the pH-dependent blockage of virus endocytosis; and (ii) the prevention of viral-specific autophagy through inhibition of the S-Phase kinase associated protein 2 (SKP2) (Figure 2). A third mechanism, that is, the direct inhibitory activity of niclosamide against ACE2 receptor has been proposed, but it requires further confirmation [163].

In subsequent works, using Vero E6 cells, it was however reported that niclosamide at 10.4 µM (IC_50_ values of 0.16) exhibited moderate virucidal activities, with a negligible reduction of viral inhibition [164]. Through molecular modeling and a virtual screening, the same authors demonstrated a strong binding affinity of the drug to M^Pro^, which was further combined with a low binding affinity to the viral Spike glycoprotein, adding a new potential MoA of niclosamide against the virus (Figure 2) [164]. More recently, niclosamide (1 µM) was reported to inhibit the Spike-induced syncytia by suppressing the activation of TMEM16 proteins, a calcium-activated ion channel and a scramblase responsible for the exposure of phosphatidylserine on the cell surface (Figure 2) [165]. These results suggested that niclosamide could be used for the prevention of pneumocyte syncytia formation in the severe form of COVID-19 [165]. These observations were further confirmed in a study in which niclosamide was able to neutralize the pH of endosomes, block viral entry and rescue the cytopathic effects upon infection in a set of different cell lines [51].

These encouraging in vitro data prompted new investigations to overcome the problem of low absorption associated with niclosamide, by developing novel formulations that enable an effective delivery of the drug to the target tissue. Interestingly, the development of a Tween 60-coated niclosamide–montmorillonite (NIC-MMT) hybrid system based on the drug carrier MMT that possesses mucoadhesive properties, resulted in an increase of the oral bioavailability of niclosamide by more than a 1.6-fold [166]. Furthermore, using Vero E6 and ACE2-expressing lung epithelial cells, a cost-effective lipid nanoparticle formulation of niclosamide (nano NCM) that employs only the FDA-approved excipient demonstrated to be active at 0.595 nM (IC_50_) with a IC_90_ of 3.38 μM, while achieving a selectivity index (CC_50_/IC_50_ ratio) of 52, whereas the non-formulated niclosamide presented selectivity index of 464. The formulation resulted quite promising, due to its stability and the possibility to be prepared in various isotonic vehicles at neutral pH [167]. Later in 2021, universal prophylactic nasal throat sprays as early treatments against SARS-CoV-2 and its more contagious variants were investigated. In particular, a low dose, prophylactic solution of niclosamide (20 μM) at a nasally safe and acceptable pH of 7.96, and a throat spray of up to 300 μM at pH 9.19 were proposed as the simplest and potentially the most effective formulations from both an efficacy as well as manufacturing and distribution standpoints, since no cold chain would be then required [168].

Recently, the activity of niclosamide against several SARS-CoV-2 variants of concern has been investigated. The results demonstrated that niclosamide inhibited the replication of the SARS-CoV-2 original strain (Wuhan D614) in VeroE6 TMPRSS2 cells with an IC_50_ of 0.13 μM and IC_90_ of 0.16 μM, in accordance with previous studies. Importantly, niclosamide also blocked the replication of the SARS-CoV-2 D614G, B.1.1.7, B.1.351 and B.1.617.2 variants with an IC_50_ of 0.06 μM, 0.08 μM, 0.07 μM, and 0.08 μM, respectively, showing similar potency across the different strains compared to the original Wuhan D614 strain [169].

#### 4.1.2. Niclosamide: Clinical Trials

A discrete number of clinical trials focused on the use of niclosamide as an effective COVID-19 treatment have been conducted in patients with different disease severities. No study results are available yet on the official website except for the study NCT04399356 (Table 3), where no significant differences in oropharyngeal clearance of SARS-CoV-2 at day 3 between placebo and niclosamide groups were reported [170]. The remaining 13 studies are summarized in Appendix A.

### 4.2. Nitazoxanide

Nitazoxanide (NTZ) or 2-(acetyloxy)-N-(5-nitro-2-thiazolyl) benzamide (Figure 1), was synthesized in the early 1970s from niclosamide [171] and approved by the FDA as a safe and effective oral antiprotozoal drug to treat intestinal parasitic infections caused by *Cryptosporidium parvum* and *Giardia intestinalis* both in adults and children [172,173]. In addition, NTZ and its active circulating metabolite tizoxanide (TIZ) also exhibit antimicrobial effects against bacteria including *Mycobacterium tuberculosis* [174,175]. In these cases, the main mechanism of action of NTZ appears to be the inhibition of pyruvate:ferredoxin oxidoreductase (crucial for anaerobic energy metabolism), along with the disruption of the microorganism’s pH homeostasis [176]. Remarkably, NTZ has shown promising activity as a broad-spectrum antiviral agent against RNA or DNA viruses in vitro, including different influenza strains, respiratory syncytial virus, parainfluenza, rotavirus, norovirus, hepatitis B and C, Dengue, yellow fever, Japanese encephalitis virus, HIV and MERS coronavirus [171,177]. Of note, both NTZ and TIZ have generally shown similar inhibitory activity against viruses in vitro [177].

In February 2020, Wang and colleagues confirmed that NTZ was able to inhibit the novel coronavirus at low micromolar concentrations (IC_50_  =  2.12 μM; CC_50_  >  35.53 μM), although the antiviral potency of the active metabolite TIZ was absent from the study [42]. In fact, only one preprint posted in December 2021 claimed that both NTZ and TIZ show similar antiviral results in Vero E6 and Caco-2 cells [178].

By April 2020, thanks to its good safety profile at approved doses along with its affordable price as a generic drug, NTZ was considered a potential treatment against COVID-19 [179]. However, concern over the unknown hepatorenal and cardiovascular effects as well as teratogenicity retarded its employment. In July 2020, based on a pathophysiological and pharmacological approach, an article complained about the “neglected” consideration of NTZ against SARS-CoV-2 infection, and recommended to initiate a clinical trial combining NTZ with AZ in early stages of COVID-19 [180]. Indeed, NTZ+AZ in mild and moderate COVID-19 patients with diarrhea (N = 20), showed a symptomatic improvement on the fifth day after treatment [181]. The same report also proposed through molecular docking that NTZ would bind to the viral ADP-ribose phosphatase domain (ADPRP or MacroD) within the non-structural nsp13 multidomain (Figure 2) [181]. Since MacroD is a key enzyme that enables the virus to bypass the host immunity [182], NTZ might provide effective antiviral therapeutics.

Considering pregnancy, in September 2020, a small prospective follow-up and report of COVID-19 cases including pregnant women (N = 20), acutely ill hospitalized (N = 5) and ambulatory patients (N = 16), found NTZ treatment to be useful against SARS-CoV-2 in an early intervention in critical conditions and during pregnancy [183]. Similarly, a cross-sectional study reported on the survival safety for COVID-19 in pregnant women treated with NTZ along with a good safety profile of the drug [184].

More recently, NTZ treatment for five days of adult patients with mild COVID-19 symptoms failed to resolve symptoms compared to placebo, although the same study also showed that early NTZ therapy was safe and significantly reduced viral load [185]. Next, the results of a randomized, double-blind pilot clinical trial comparing NTZ vs placebo for seven days, linked NTZ with improved clinical (reduced hospitalization), virologic (faster negative tests by qPCR) and lower inflammatory outcomes (ID NCT04348409) [186]. Moreover, the combination of NTZ, ribavirin and IVM plus Zinc compared to routine supportive treatment against COVID-19, led to a faster viral clearance from the nasopharynx than symptomatic therapy [187] (ID NCT04392427). Finally, evidence that oral NTZ may reduce the risk of severe illness for patients at high risk, and the time to sustained recovery for mild COVID-19 individuals has been reported (ID NCT04486313) [188].

From the mechanistic point of view, NTZ seems to impair SARS-CoV-2 Spike protein variants maturation and to induce the generation of viral protein modifications, hindering progeny virion infectivity and Spike-mediated pulmonary cell fusion (Figure 2) [189], which highlights the potential of NTZ to fight against SARS-CoV-2 infection.

As an immunomodulator, NTZ was able to suppress the production of IL-6 in vitro and in an animal model injected with lipopolysaccharide and thioglycolate [190], and to inhibit the production of other cytokines in peripheral blood mononuclear cells [191]. These results have been recently reproduced in COVID-19 patients (Figure 2) [186]. In addition, NTZ amplifies the pathways of type 1 interferons (IFN-Is) (Figure 2) [171,177], which are found quite low during severe COVID-19 immunopathology [192]. Importantly, in 2019, NTZ was proved to be a potent bronchodilator even under harsh conditions using maximally contracted airways or airways pretreated with a cytokine cocktail [193]. These findings might be of critical importance for COVID-19 patients, since NTZ might alleviate respiratory symptoms including in mechanically ventilated patients.

### 4.3. Ivermectin

Ivermectin (IVM) is a macrocyclic lactone made as an 80:20 mixture of two homologues, 22,23-dihydroavermectin B1a and 22,23-dihydroavermectin B1b, respectively [194]. It belongs to the avermectin family of natural compounds produced by the bacterium Streptomyces avermitilis, and was discovered as an anthelminthic agent in 1979 to treat various parasitic infections [195,196].

Although different MoA are reported for IVM, one of its main cellular activities involve the targeting of the importin alpha (IMPα) and the inhibition of normal nuclear transport [197], required for normal transcription and DNA replication [198]. IVM binds to IMPα, which prevents IMPα interaction with IMPβ and the subsequent recognition of a specific nuclear localization signal (NLS) on the cargo, thus blocking the nuclear transport.

In the last 10 years, IVM has been identified as an antiviral agent, both against RNA viruses such as HIV-1, influenza or flaviviruses, and against DNA viruses such as polyomaviruses and adenoviruses [197,199], by suppressing viral replication/maturation through the inhibition of viral proteins to be transported into and out of the nucleus [197,200]. Interestingly, the anti-SARS-CoV-2 effects of IVM seem to be linked to the inhibition of nuclear transport of viral proteins as well [201,202,203,204].

#### 4.3.1. Ivermectin: In Silico and In Vitro Studies

During the first wave of the pandemic in Europe and America, the FDA approved the repurposing of IVM for the treatment of COVID-19. Strikingly, it was early reported that treating SARS-CoV-2-infected Vero-hSLAM cells with 5 µM of IVM 2 h post infection, induced a ~5000-fold reduction of viral RNA both inside the cells and in the cell supernatants, with an antiviral activity lasting for 48 h with no toxic effects [204].

To confirm the efficacy of IVM on SARS-CoV-2 replication, an in silico study conducted in June 2020 demonstrated that IVM may interfere with the virus attachment to the host’s ACE2 receptor (Figure 2) [205]. It was later confirmed that the 22,23-dihydroavermectin B1b homologue was able to interact with the ACE2′s RBD domain [206]. Through stable isotope labeling amino acids in cell culture (SILAC) quantitative proteomics, a wide-spectrum of antiviral effects of IVM was reported, since as much as 52 SARS-CoV-2 proteins were found to be altered after treatment with IVM [207]. In particular, it was found that IVM impairs viral replication through direct inhibition of the enzymatic activity of M^Pro^ [208,209,210,211] (Figure 2).

In vitro cell-based models cannot represent the in vivo dynamics of infection in the lung microenvironment mainly due to the lack of the corresponding immune response. To this regard, a modelling study that considered the potential immune responses, the dose and the time of administration of different drugs, identified IVM as the better drug with a better efficacy when administered immediately after disease positivity [212]. In contrast, a study conducted on Calu-3 cells that better represent the lung compartment, failed to report a reduction of SARS-CoV-2 viral load in the presence of a high IVM dose [213].

At the end of March 2021, other two in silico reports showed high binding affinities of IVM for the viral replicase, protease and RBD of SARS-CoV-2, and the human TMPRSS2 receptor (Figure 2), pushing the use of IVM as a candidate therapy against COVID-19 [214,215]. More recently, Gonzalez-Paz and colleagues evaluated for the first time the effectiveness of each homolog comprising IVM. With an elastic networks model and computational and biophysical approaches, the authors observed that 22,23-dihydroavermectin B1a exhibits high affinity for the IMPα and IMPβ subunits (Figure 2), while 22,23-dihydroavermectin B1b presents higher affinity for viral structures [216,217], especially with the SARS-CoV-2 RBD domain, further speculating that it might block the ACE2-RBD binding [206].

To increase efficacy and reduce the risk of resistance emergence, IVM was also tested in combination with other drugs. Indeed, an in vitro study on Vero E6 cells successfully demonstrated that IVM in combination with remdesivir enhanced their single antiviral activities, although it was warned that different routes of administration may modify the ability of the drugs to simultaneously reach therapeutic concentrations [218]. Similar conclusions were reached by Eweas et al. through molecular docking assays [219]. Lastly, an in vitro model of infection combined with confocal microscopy investigated the anti-SARS-CoV-2 effects of IVM in combination with atorvastatin (ATV), observing that both drugs blocked the activation of the NF-κB signaling pathway, modified gene expression of IMPα and Rho GTPase (targets of IVM and ATV, respectively), and finally suppressed IMPα accumulation in the nucleus, presenting both IVM and ATV as valid drugs against the virus [220].

#### 4.3.2. Ivermectin: Preclinical Studies

An indirect proof of the in vivo efficacy of IVM against the novel coronavirus was first reported in November 2020, since IVM treatment succeeded in reducing the liver viral load of mouse hepatitis virus (MHV) in the murine model, a single-stranded RNA virus like SARS-CoV-2 [221]. Other important in vivo study in SARS-CoV-2 infected hamsters showed that despite no effects in the reduction of viral load in the respiratory tracts, IVM reduced the IL-6 and IL-10 ratio and induced the polarization of macrophages towards a M2 phenotype that favored a beneficial anti-inflammatory response, inferring a potential positive implication of IVM in the clinical condition of COVID-19 patients (Figure 2) [222].

#### 4.3.3. Ivermectin: Retrospective, Observational Studies, and Clinical Trials

With such promising in silico, in vitro, and in vivo data, many clinical trials, observational trials, or retrospective studies, mainly conducted in Southern Hemisphere countries, have been performed to evaluate the potential role of IVM against COVID-19.

##### IVM Used as Monotherapy

Concerning the safety of IVM administration, old studies also reported a good tolerability of the drug at doses higher than those approved by the FDA for human use, and with repeated drug administrations, exhibiting no toxicity for the central nervous system [223], and during pregnancy [224]. However, in vivo studies showed that IVM may induce malformation in the fetus and, moreover, the combination with other drugs would increase the risk of such malformations [225]. It remains to be determined if IVM is safe in COVID-19 pregnant woman, mainly due to exclusion of pregnant woman in almost all trials performed so far [226].

Compared to in vitro studies, human pharmacokinetic analyses demonstrated that IVM would achieve lung concentrations over 10 times higher than the reported IC_50_ [227]. Interestingly, a significant early seronegativity was observed in patients receiving IVM for five days compared to placebo and the IVM-doxycycline groups in Bangladesh [228], a result that was further confirmed by African and Brazilian trials which reported a reduction in the viral load in a dose-dependent manner [229,230]. In addition, an IVM-mucoadhesive nanosuspension nasal spray showed effectiveness in mild COVID-19 patients with a faster viral clearance and a faster resolution of anosmia [231]. Despite these encouraging findings, by conducting a survey of the clinical data collected so far, there is a certain agreement that a single dose of IVM does not significantly reduce viral load. In contrast, if given early after the onset of symptoms, it helps recovering faster from anosmia and hyposmia, it prevents the progression of the disease and it improves the clinical outcome compared to untreated patients [232,233,234,235,236,237].

In different studies, however, no beneficial effects of IVM on mild or severe disease were observed, as the recovery time and the resolution of symptoms were not significantly different from untreated patients [238,239,240,241]. Contrasting results have been reported as well. An Indian clinical trial conducted by Shahbaznejad and colleagues showed that important clinical COVID-19 symptoms such as dyspnea, cough, and lymphopenia, were improved in patients treated with a single dose of IVM [242]. Even more, the administration of IVM within the first 3 days after recovery in an intensive care unit improved gastrointestinal symptoms and the number of ventilator-free days in COVID-19 patients affected by severe syndrome and subjected to ventilation [243]. Contrarily to these observations, no improvement in clinical and virological outcomes were observed among the IVM-treated compared to patients treated only with the SoC [244]. Regarding the hospitalization parameters, a reduced hospitalization trend in IVM-treated patients has been reported [245], whereas other studies did not observe such decrease in mild-moderate COVID-19 subjects [246,247]. Finally, the role of IVM in the recovery from long-COVID-19 was investigated in a retrospective study conducted in Argentina, showing that IVM induces a faster reduction of the symptoms [248].

With regard to mortality, IVM treatment seems to be associated with lower mortality rates in patients with severe pulmonary damage [249]. In a multi-center clinical trial, the mortality rate decreased by 15% only in hospitalized individuals receiving different doses of IVM and different administration routes, while those receiving HCQ did not [250]. Besides, an Egyptian study observed a decreased in IL-6 and IL-1 cytokines in COVID-19 patients treated with IVM, with a reduction in the mortality rate and the hospitalization time [251]. Conversely, studies conducted in Pakistan and China did not detect differences in mortality rate, inflammatory markers, time of hospitalization, time of resolution of symptoms and syndrome course in mild and severe COVID-19 subjects [252,253]. Likewise, an American prospective study on 286 patients treated with two doses of IVM found no benefits in hospitalization time, intensive care unit admission, intubation rate or mortality [254].

##### IVM Treatment in Combination with Other Drugs

To improve the effectiveness of IVM, the repurposing of this drug in combination with other drugs has been extensively tested. The first three studies were conducted in Bangladesh in the summer of 2020, demonstrating that mild-moderate COVID-19 patients receiving IVM in combination with doxycycline showed a faster negative conversion in PCR and significantly faster symptom resolution compared to the HCQ-AZ-treated group [255]. Moreover, IVM-doxycycline also resulted in a faster viral clearance [256] in severe COVID-19 patients [257], together with improvements in early recovery and prevention to progression to more serious disease [258]. In addition, in a case series of SARS-CoV-2 positive patients hospitalized in Pakistan, IVM combined with doxycycline, remdesivir, tocilizumab, enoxaparin sodium and steroids, emerged as potential new treatment options by reducing the disease-associated severity and recovery rate [259].

Besides, a retrospective case series performed in India in June 2020, evaluating the effect of IVM combined with atorvastatin or N-acetylcysteine, showed that 98.6% of patients successfully recovered, with a mortality rate lower than the national rate [260]. Furthermore, IVM in combination with AZ compared with a single IVM dose was more effective in reducing the duration of the symptoms [261]. Additionally, a small retrospective study upon a 10-day treatment schedule composed of IVM-AZ-Cholecalciferol enabled a significant decrease in the recovery time at early stages of COVID-19 [262]. Additionally, positive effects in preventing hospitalization and death were observed in Mexico, after a therapy with IVM, AZ, Montelukast, and acetylsalicylic acid [263]. Further, the combination of IVM, AZ, acetaminophen, and aspirin, given to mild COVID-19 patients with no respiratory failure, reduced the risk of hospitalization and mortality [264]. On the contrary, the combination of IVM and HCQ did not reduce the viral load nor the mortality rate [265,266].

Beyond dual combination therapies, a triple combination of IVM-HCQ-AZ resulted in a faster negative conversion compared to AZ-HCQ-treated patients [267], a positive effect on clinical recovery, reduction of positivity time, and community spread prevention [268], and an overall improvement of clinical parameters [269]. Moreover, different mixed therapies of IVM with nitazoxanide, ribavirin, and zinc supplement, led to earlier negative conversions in affected patients compared to the standard care therapy [187]. Finally, to prevent COVID-19-associated complications such as hypercoagulation, the combination of a higher approved dose of IVM with dexamethasone, enoxaparin injection and aspirin (I.D.E.A protocol) resulted in lower mortality rates and no hospitalization among mild COVID-19 patients [270].

##### Prophylactic Use of IVM

To avoid the spreading of the virus and to block its transmission, the prophylactic use of IVM has been investigated especially in healthcare workers, who are highly exposed to the risk of infection. During summer 2020, a protective role of IVM with very low and mild adverse effects was firstly reported: only 7.4% of asymptomatic contacts of SARS-CoV-2 positive patients that received IVM developed COVID-19 vs 58.4% of untreated subjects [271]. Preventive IVM administration was also tested on healthcare workers in India and in the Dominican Republic, with a reduction of the viral infection by 73% and 70%, respectively [271,272,273]. Consistent with these results, other observational studies found a positive involvement of IVM in preventing SARS-CoV-2 infection, since a single oral monthly dose for 4 months reduced the viral infection by ten-fold [274]. Moreover, two prophylactic doses of IVM in the following months after COVID-19 infection showed better results compared to a single IVM dose [275]. Even more, prophylactic IVM in combination with Iota-Carrageenan (a sulphated polysaccharide extracted from red seaweeds), suggested a preventive role of this combination therapy [276,277]. On the other hand, no prevention of viral infection was reported by a clinical trial conducted in Singapore, performed on 3037 COVID-19 negative migrant workers subjected to quarantine who were treated with prophylactic drug regimens, including IVM [278]. In Africa, remarkably, where IVM is massively administrated to combat non-related parasitic diseases, the reduced COVID-19 positive rates and deaths might be explained by prophylactic IVM use against SARS-CoV-2 infection [279,280].

##### Ivermectin: Summary

To date, 148 studies, 98 peer-reviewed, and 78 with results have been performed on the use of IVM against SARS-CoV-2 [281]. These studies present many limitations, including the small cohorts of enrolled patients, the different posology and the different disease stages at which IVM efficacy has been investigated. IVM has been officially adopted as an early COVID-19 treatment for 28% of the world population, especially in Indonesia, Bangladesh, Mexico, Egypt, Ukraine, and Venezuela [281]. Since other drugs have been concomitantly administrated with IVM, it is difficult to draw a clear conclusion about the real anti-SARS-CoV-2 activity of this antiparasitic drug. Moreover, no optimal dose has been established so far [282]. Even if some clinical trials are still ongoing (Appendix A Appendix A), the use of IVM as a legitimate therapeutic option to combat the novel SARS-CoV-2 infection has not yet been authorized or approved by the FDA, predominantly due to the absence of sufficient supportive data [283]. Table 4 contains available results from observational and clinical trials on the use of IVM as anti-COVID-19 therapeutics.

## 5. Conclusions

Over the past two years, great efforts have been made to find an effective treatment for COVID-19, and the use of antiparasitic drugs rapidly appeared as one of the most immediate solutions, based on their previously reported antiviral activities. Unfortunately, despite the promising in vitro results, the anti-SARS-CoV-2 activity of CQ and HCQ failed to be translated into clinical efficacy, as documented by several reviews and the randomized clinical trials [287]. In the case of CQ, disparity between laboratory and clinical experiments could be mainly attributed to complex pharmacokinetics, specifically its distribution time into different tissues and its prolonged elimination half-life, which has made it difficult to extrapolate drug concentrations in culture media to human doses. Prophylactic activity of HCQ was explored as well, although dosing errors and the lack of consideration to patients with preexisting conditions were shown to be present in many reported trials. It is important to note that the use of HCQ as a prophylactic drug for COVID-19 must unambiguously include an extensive investigation of its adverse events, even the mild ones, such as the observed gastrointestinal disorders, along with the costs and hassle of undergoing the administration of a preventive medication. Currently, the potential benefits associated with this prophylactic use do not overcome the arising problematic issues. Based on these considerations, we feel quite confident affirming that HCQ would not be considered as a possible alternative to the ongoing vaccination campaign to prevent COVID-19.

A fundamental support to improve and accelerate the drug repurposing strategy might derive from the use of Artificial Intelligence (AI), a powerful approach that has been widely exploited in the sequencing of SARS-CoV-2 genome from the beginning of the pandemic (reviewed in [288]). In brief, AI consists of creating synergy between the structure-based, the ligand-based screening methods and the AI algorithms, to generate accurate prediction models. A clear example has been the identification of the anti-SARS-CoV-2 properties of remdevisir, atazanavir and efavirenz, which emerged in a deep learning-based drug-target interaction prediction model, known as “Molecule Transformer-Drug Target Interaction (MT-DTI) [289]. Another subdiscipline of AI is Machine Learning (ML), which able to reveal the connections between drugs, viral and host proteins, and was successfully applied to identify Baricitinib and Rifampicin as COVID-19 treatments [290]. However, despite the many advantages of AI-based methods in drug repurposing, there is a weakness that needs to be considered: AI requires large datasets, which are not always easily accessible, structured and standardized. In addition, all the candidate drugs identified by AI need to be equally validated through classic in vitro methods.

The gap between in vitro anti-SARS-CoV-2 evidence and the transferability to clinical practice was even wider when considering ivermectin. Although the failure to repurpose antiparasitic drugs against COVID-19 can be a cause for disappointment and discouragement, on the contrary, it is necessary to draw lessons to be able to carefully define potential new treatments. As stated by Ho and colleagues in a recent review [291], future studies aiming to repurpose well-known drugs for fighting emerging infections should be based on a more accurate design of the in vitro antiviral studies, as well as on improving the validation of viral targets and on in-depth studies of the translatability from the laboratory to the clinic.

Thus, as often occurs in the biological and pharmaceutical fields, innovation is widely encouraged and must be strictly integrated with the more classic methodologies, to balance benefits and costs and to obtain solid and reliable results in the fastest way possible.

## Figures and Tables

**Figure 1 microorganisms-10-01284-f001:**
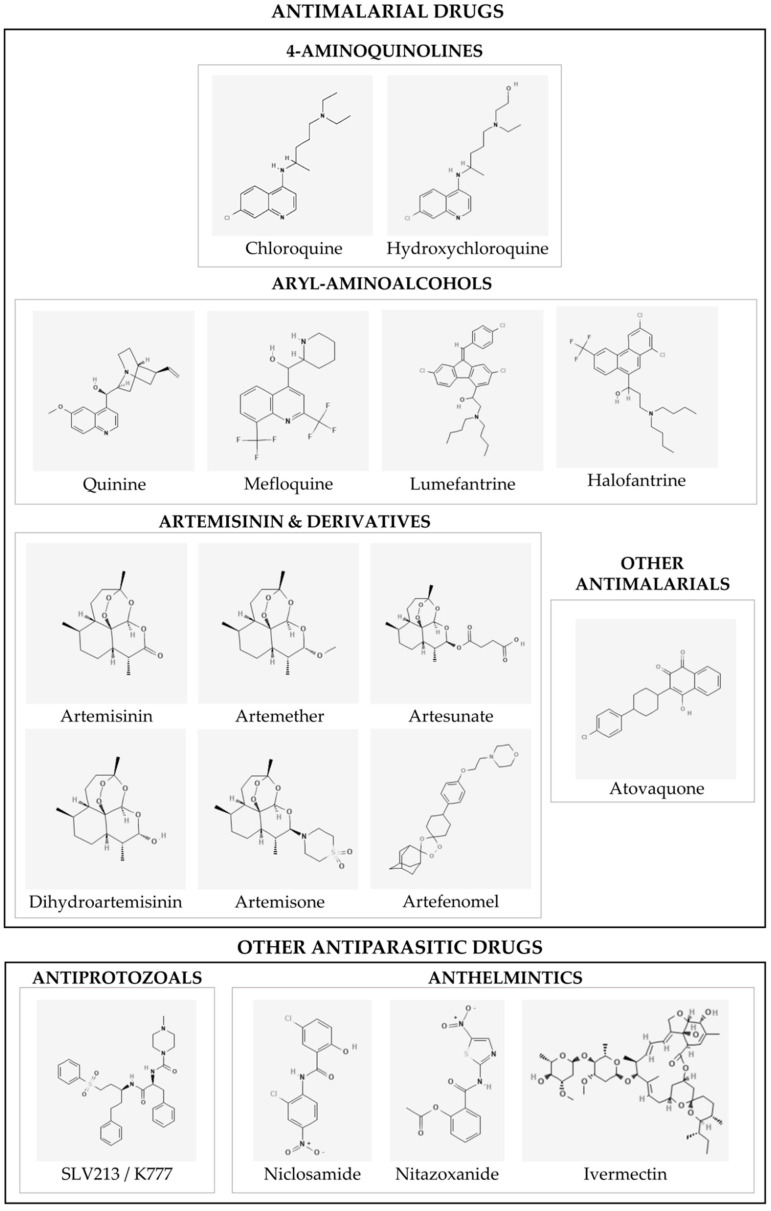
Chemical structures of antiparasitic drugs reported to have anti-SARS-CoV-2 activity and included in this manuscript. Source: PubChem.

**Figure 2 microorganisms-10-01284-f002:**
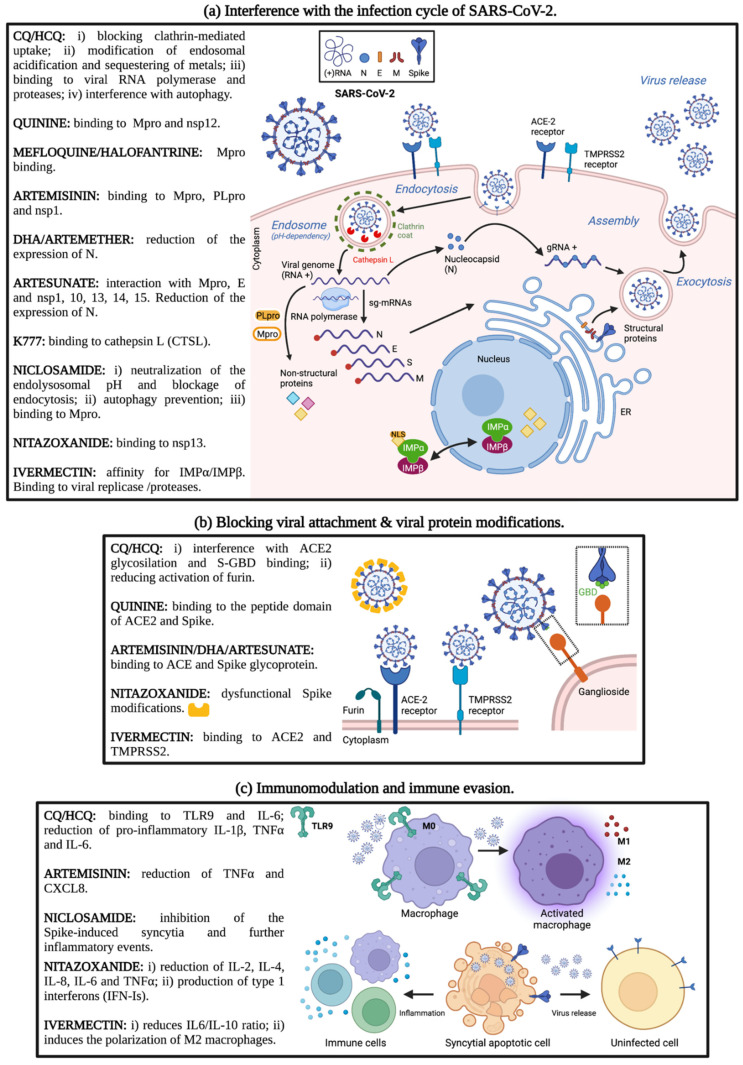
Mechanisms of action of antiparasitic drugs to interfere with SARS-CoV-2 viral infection at different levels: (**a**) during the infection cycle of the virus; (**b**) during the viral attachment to cellular receptors; and (**c**) as indirect immunomodulatory agents. Created with BioRender.com.

**Table 1 microorganisms-10-01284-t001:** CQ and HCQ clinical trials against COVID-19 with published results.

Chloroquine (CQ)				
Trial No.	Phase	Drugs	No. Participants	Status & Results
NCT04323527	Phase 2	CQ diphosphate: high dosage (600 mg twice daily for 10 days) vs. low-dosage (450 mg twice on day 1 and once daily for 4 days)	278	Completed. Higher CQ dosage should not be recommended for critically ill patients with COVID-19 because of its potential safety hazards, especially when taken concurrently with azithromycin and oseltamivir [76].
NCT04420247	Phase 3	CQ/HCQ added to standard of care (SoC)	142	Completed. The trial was stopped before reaching the planned sample size due to harmful effects. In patients with severe COVID-19, the use of CQ/HCQ added to SoC resulted in a significant worsening of clinical status, an increased risk of renal dysfunction and an increased need for invasive mechanical ventilation [99].
**Hydroxychloroquine (HCQ)**				
**Trial No.**	**Phase**	**Drugs**	**No. Participants**	**Status & Results**
NCT04381936 “RECOVERY Trial”	Phase 2/Phase 3	HCQ vs. SoC	4716	Completed (HCQ arm). COVID-19 patients receiving HCQ did not have a lower incidence of death at 28 days than those who received usual care [78].
NCT04315948 “SOLIDARITY Trial”	Phase 3	HCQ vs. Remdesivir vs. Lopinavir/ritonavir vs. Interferon Beta vs. SoC vs. AZD7442 vs. Placebo	2416	Recruiting. HCQ has little or no effect on hospitalized COVID-19 patients according to mortality, initiation of ventilation and duration of hospital stay [84].
NCT04308668	Phase 3	HCQ vs. Placebo	1312	Completed. HCQ failed to prevent illness compatible with COVID-19 or confirmed infection when used as high-risk or moderate-risk postexposure prophylaxis within 4 days after exposure [80].
NCT04304053	Phase 3	HCQ as prophylactic treatment	2300	Completed. Postexposure therapy with HCQ did not prevent SARS-CoV-2 infection or symptomatic COVID-19 in healthy persons exposed to a PCR-positive case patient [81].
NCT04332991	Phase 3	HCQ vs. Placebo	479	Completed. Among adults hospitalized with respiratory illness from COVID-19, treatment with HCQ did not significantly improve clinical status at day 14 [100].
NCT04466540	Phase 4	HCQ vs. Placebo	1372	Completed. HCQ did not reduce the risk of hospitalization in outpatients with mild or moderate forms of COVID-19 [89].
NCT04321278	Phase 3	HCQ vs. HCQ + azithromycin	447	Completed. In patients with severe COVID-19, the use of HCQ + azithromycin did not improve clinical outcomes [101].
NCT04325893	Phase 3	HCQ vs. Placebo	259	Terminated (decrease in number of eligible patients). Trial involving mainly older patients with mild to moderate COVID-19. HCQ treatment did not result in better clinical or virological outcomes [88].
NCT04403100	Phase 3	HCQ Sulfate Tablets vs. Lopinavir/Ritonavir Oral Tablet vs. HCQ Sulfate Tablets +Lopinavir/Ritonavir Oral vs. Placebo	1968	Recruiting. Neither HCQ nor lopinavir-ritonavir showed any significant benefit for decreasing COVID-19–associated hospitalization or other secondary clinical outcomes [86].
NCT04354428	Phase 2/Phase 3	HCQ sulfate vs. HCQ+AZ	300	Active, not recruiting. HCQ and HCQ + AZ do not affect the clinical course of COVID-19 among outpatients and should not be used to treat SARS-CoV-2 infection [85].

**Table 2 microorganisms-10-01284-t002:** Artemisinin clinical trials with known results.

**Trial No.**	**Phase**	**Drugs**	**No. Participants**	**Status & Results**
NCT05004753	Phase 4	ARTIVeda™ (Artemisinin) +/− SoC	120	Completed. ARTIVeda™ provides a faster recovery of patients with mild-moderate COVID-19 (preliminary data on 60 patients) [136]

**Table 3 microorganisms-10-01284-t003:** Clinical trials involving niclosamide as antiviral agent for COVID-19 patients’ treatment with published results.

Trial No.	Phase	Drugs	No. Participants	Status & Results
NCT04399356	Phase 2	Niclosamide vs. Placebo	73	Completed. No significant difference in oropharyngeal clearance of SARS-CoV-2 at day 3 between placebo and niclosamide-treated groups [170].

**Table 4 microorganisms-10-01284-t004:** IVM: observational and clinical studies against COVID-19.

Observational Studies				
Trial No.	Phase	Drugs	No. Participants	Status & Results
**NCT04434144**	NA	IVM + Doxycycline/HCQ + AZ	116	Completed. Faster negative conversion of PCR and significantly faster symptom resolution in IVM-Doxycycline [255].
**NCT04425863**	NA	IVM + Aspirin/ IVM + Dexamethasone injection + Aspirin/ IVM + Dexamethasone injection + Enoxaparin injection	167	Completed. A positive role of IVM + aspirin + dexamethasone + enoxaparin therapy. No hospitalization of mild cases and lower mortality rate compared to national rate [270].
**Clinical Trials**				
**Trial No.**	**Phase**	**Drugs**	**No. Participants**	**Status & Results**
**NCT04381884**	2	IVM and SoC vs. SoC	45	Completed. No differences between the two groups in viral load and in clinical evolution of the patients. Significant difference between patients that showed high IVM levels in plasma samples vs. untreated patients: correlation of IVM level with decrease of the viral load. High dose of IVM did not show toxicity [233].
**NCT04591600**	1/2	IVM + Doxycycline/SoC	140	Completed. Therapy improvement by IVM + Doxycycline.
**NCT04343092**	1	IVM vs. IVM + HCQ + AZT	16	Completed. Better efficacy of combination of IVM, HCQ and AZT; shorter hospitalization, and safety.
**NCT04405843**	2/3	IVM vs. Placebo	476	Completed. No significant resolution of symptoms [284].
**NCT04529525**	2/3	IVM vs. Placebo	501	Completed. No effect on preventing hospitalization of COVID-19 patients [247].
**NCT04391127**	3	IVM vs. Placebo vs. HCQ	108	Completed. No efficacy of IVM or HCQ in decreasing hospitalization days, respiratory problems or deaths.
**NCT04523831**	3	IVM + Doxycycline vs. SoC	400	Completed. Improvements in earlier recovery, prevention to progress to more serious disease (mortality) and increased likeliness to be COVID-19 negative by RT-PCR [258].
**NCT04784481**	1/2	IVM	254	Completed. Reduction of the number of symptoms and improvement of clinical state.
**NCT04701710**	1/2	IVM + Iota-carrageenan	300	Completed. Reduction of number of infected health workers with preventive treatment with IVM and Iotacarrigean. Prevention of severe disease.
**NCT04390022**	1/2	IVM vs. Placebo	24	Completed. No difference between treated and untreated patients in the decrease of viral load but early recovery of anosmia and hyposmia in IVM-treated patients [234].
**NCT04422561**	2/3	IVM chemoprophylaxis vs. no treatment	304	Completed. Significant differences among the two groups: 7.4% SARS-CoV-2 positive in IVM-treated vs. 58.4% of untreated subjects. Protective role of IVM [271].
**NCT04446104**	3	IVM vs. HCQ vs. Zinc vs. Povidone-Iodine vs. Vitamin C	4257	Completed. No efficacy of IVM [278].
**NCT04438850**	2	IVM vs. Placebo	93	Completed. Incidence dramatically dropped and is lack of eligible patients.
**NCT04602507**	2	IVM vs. Placebo	75	Completed. Lack of severe COVID-19 cases in the place of study.
**NCT04374019**	2	IVM vs. Camostat Mesilate vs. Artemesia annua vs. AS	13	Completed. Slow accrual.
**NCT04431466**	2	IVM vs. SoC	32	Completed. IVM for SARS-CoV-2 treatment is safe. IVM antiviral effect is dose-dependent [230].
**NCT04716569**	2/3	Intranasal IVM spray	150	Completed. Reduction of anosmia and rapid viral clearance in treated patients [231].
**NCT04779047**	4	IVM vs. HCQ vs. Remdesivir vs. Tocilizumab vs. Lopinavir/Ritonavir 150	150	Completed. No positive effect of IVM treatment [285].
**NCT04482686**	1	IVM + Doxycycline + Zinc + Vitamin D3 + Vitamin C	31	Completed. The combination of drugs is safe and effective [286].

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
