# Peer review of "Antiparasitic Drugs against SARS-CoV-2: A Comprehensive Literature Survey"

_microorganisms, 2022, doi:10.3390/microorganisms10071284_

Round 1

Author Response

Attached file.

Reviewer 2 Report

This is a thorough review of the available literature regarding the re-purposing of antiparasitic drugs against SARS-CoV-2. Because of its length, it is unlikely to be read in its entirety by viewers. Would suggest shortening the article (by at least half) to include relevant referencing to the points being made by the authors.

Suggest deleting "old" and "a window of opportunity" from the title

If included in the revised manuscript, need a reference on line 1008.

On line 1198, the authors mean re-purpose, not re-proposal

Author Response

Attached file.
